

# Endogenous testosterone correlates with parochial altruism in relation to costly punishment in different social settings

Luise Reimers[*], Eli Kappo, Lucas Stadler, Mostafa Yaqubi and Esther K. Diekhof[*]

Faculty of Mathematics, Informatics and Natural Sciences, Department of Biology, Institute of Zoology, Neuroendocrinology and Human Biology Unit, Universität Hamburg, Hamburg, Germany
[*] These authors contributed equally to this work.

Corresponding author
Esther K. Diekhof,
esther.diekhof@uni-hamburg.de

## ABSTRACT

Testosterone plays a key role in shaping human social behavior. Recent findings have linked testosterone to altruistic behavior in economic decision tasks depending on group membership and intergroup competition. The preferential treatment of ingroup members, while aggression and discrimination is directed towards outgroup members, has been referred to as parochial altruism. Here we investigated in two consecutive studies, whether testosterone is associated with parochial altruism depending on individual tendency for costly punishment. In the first study, 61 men performed a single-shot ultimatum game (UG) in a minimal group context, in which they interacted with members of an ingroup and outgroup. In the second study, 34 men performed a single-shot UG in a more realistic group context, in which they responded to the proposals of supporters of six political parties during the German election year 2017. Political parties varied in their social distance to the participants' favorite party as indicated by an individual ranking. Participants of study 2 also performed a cued recall task, in which they had to decide whether they had already encountered a face during the previous UG (old-new decision). In order to make the UG data of study 2 most comparable to the data of study 1, the rejection rates of several parties were combined according to the social distance ranking they achieved. Parties ranked 1 to 3 formed the relatively close and favored 'ingroup' that shared similar political values with the participant (e.g., left wing parties), while the 'outgroup' consisted of parties ranked from 4 to 6 with more distant or even antagonistic political views (e.g., conservative to right wing parties). In both studies, results showed a parochial pattern with higher rejection rates made in response to outgroup compared to ingroup offers. Interestingly, across studies higher salivary testosterone was associated with higher rejection rates related to unfair outgroup offers in comparison to the unfair offers made by ingroup members. The present findings suggest that latent intergroup biases during decision-making may be positively related to endogenous testosterone. Similar to previous evidence that already indicated a role of testosterone in shaping male parochial altruism in male soccer fans, these data underscore the general, yet rather subtle role of male testosterone also in other social settings.

## INTRODUCTION

Humans display a large extent of prosocial behaviors such as cooperation and altruism. At the same time, the human history of conflicts and wars is unparalleled. This supposedly inconsistent behavior of ingroup favoritism and outgroup hostility has recently been referred to as *parochial altruism* (*Choi & Bowles, 2007*). A behavior is thereby defined as altruistic if it incurs personal costs without direct benefits or only minimal benefits for the actor, but happens to benefit another person or a group of people (in case of parochial altruism this would be the ingroup). For example, in order to protect members of the ingroup against outgroup threat individuals may have to engage in hostile acts with outsiders, and are willing to do so, despite negative consequences like death or mutilation. Empirical evidence for parochial altruism comes from several studies that used economic decision tasks to emulate real-world situations in which limited resources are unequally distributed between competing human groups (*Baumgartner et al., 2012*; *Bernhard, Fischbacher & Fehr, 2006*; *Diekhof, Wittmer & Reimers, 2014*; *Fershtman & Gneezy, 2001*; *Goette et al., 2012*; *Kubota et al., 2013*; *Reimers & Diekhof, 2015*). For instance, members of different indigenous language groups in Papua New Guinea have been shown to display a strong ingroup bias by punishing norm violators that treated members of their own group unfairly more often compared to situations, in which the "victim" of the unequal share was an outgroup member (*Bernhard, Fischbacher & Fehr, 2006*). In addition, this ingroup bias seemed to be accompanied by a stronger tendency towards outgroup hostility in contexts that involved a competition between groups. Accordingly, in such a group competition context, army platoon members punished members of other platoons more harshly than in a neutral decision context, and even if the outgroup members were cooperative (*Goette et al., 2012*). It thus appears as if humans have a tendency for both ingroup favoritism and outgroup hostility that may be explained with the prevalent intergroup conflicts in human ancestry. Since cooperation within the own group and successfully competing against outgroups was crucial in terms of survival, these conflicts have been proposed to have led to the evolution of both altruism and parochialism (*Bowles, 2009*; *Choi & Bowles, 2007*).

Accumulating evidence has previously linked testosterone to economic decision making in social interactions, yet studies mostly revealed inconsistent results. For instance, some studies found that testosterone was associated with increased fairness preference and thus higher rejection rates in response to unfair proposals made in the context of an ultimatum game (UG) (*Burnham, 2007*; *Dreher et al., 2016*; *Eisenegger et al., 2010*; *Mehta & Beer, 2010*; *Diekhof, Wittmer & Reimers, 2014*), which has been interpreted as an act of altruistic punishment as it involves the loss of the offered share. Others found the opposite, namely that testosterone administration may be related to increased greediness and selfishness as well as a reduced fairness preference in the UG, which was reflected by either reduced punishment of violations of the fairness norm or reduced generosity when being in the role of the UG proposer (*Kopsida et al., 2016*; *Zak et al., 2009*). Finally, some studies found no effect at all (*Cueva et al., 2017*; *Zethraeus et al., 2009*). However, the above-mentioned studies differed in their methodological approach: some investigated endogenous testosterone effects (*Burnham, 2007*; *Mehta & Beer, 2010*; *Diekhof, Wittmer*

& Reimers, 2014), while others tested the effect of testosterone administration (*Cueva et al., 2017*; *Eisenegger et al., 2010*; *Kopsida et al., 2016*; *Zak et al., 2009*; *Zethraeus et al., 2009*). Moreover, samples consisted either of men (*Burnham, 2007*; *Cueva et al., 2017*; *Dreher et al., 2016*; *Zak et al., 2009*; *Diekhof, Wittmer & Reimers, 2014*), or women (*Eisenegger et al., 2010*; *Zethraeus et al., 2009*) or were mixed (*Kopsida et al., 2016*; *Mehta & Beer, 2010*). This made it difficult to discern the actual effect of testosterone on economic decisions in the UG, and well-designed replication studies are currently lacking.

As to the mechanism underlying parochial altruism, testosterone has recently been proposed to be an important mediator of decision-making in the UG and related economic decision tasks that involved an intergroup factor (*Diekhof, Wittmer & Reimers, 2014*; *Reimers, Büchel & Diekhof, 2017*; *Reimers & Diekhof, 2015*). One recent study investigated the behavioral effects of endogenous testosterone by accounting for group membership and intergroup competition. Male soccer fans played the UG against other soccer fans of either their own favorite team (ingroup) or of other teams of varying enmity and social distance (outgroups) (*Diekhof, Wittmer & Reimers, 2014*). In the UG, two players bargain about how to split an initial endowment (*Güth, Schmittberger & Schwarze, 1982*). Soccer fans with high testosterone levels offered more points to the ingroup and also rejected rather fair offers (40% of initial endowment), when these were made by an outgroup in a competitive relative to a neutral decision context, in which an additional group reward could be acquired (*Diekhof, Wittmer & Reimers, 2014*). A similar pattern of increased ingroup cooperation in the face of intensified intergroup competition was found in a prisoner's dilemma, another economic decision task measuring cooperation rates (*Reimers & Diekhof, 2015*). Moreover, a recent neuroimaging study has provided first evidence for testosterone's action in the brain in a sample of male soccer fans playing the UG (*Reimers, Büchel & Diekhof, 2017*). The results indicate dissociable testosterone-brain correlations depending on individual tendency for costly punishment. In individuals with a more selfish (i.e., economically rational) strategy and lower rejection rates in response to unfair offers of ingroup members, testosterone was positively correlated with activity in the anterior insula. In the context of the UG, the anterior insula has been implicated in processing of negative emotions and norm violations (*Civai et al., 2012*; *Sanfey et al., 2003*), as well as with positive emotional affect in other contexts (e.g., *Hennenlotter et al., 2005*). Therefore, the positive association between testosterone and insular activation despite lower rejection rates could be interpreted as the voluntary decision against sanctioning an unfair ingroup member, thus supposedly reflecting increased ingroup favoritism in a social dilemma situation (*Reimers, Büchel & Diekhof, 2017*). In inequity averse individuals (i.e., subjects with strong fairness preference and generally enhanced rejection rates) high testosterone was predictive of increased activity in ventromedial prefrontal regions, which have previously been associated with monitoring of subjective reward value (*Amodio & Frith, 2006*; *Plassmann et al., 2008*). Despite similar rejection rates for ingroup and outgroup offers, this was observed in response to unfair outgroup proposers only, thus supposedly reflecting the increased reward value of sanctioning a norm-violating outgroup member. Interestingly, the selfish players also showed a stronger tendency towards parochial altruism (higher rejection rates in response to unfair outgroup than ingroup offers) even in absence of an intergroup

competition. Based on these observations we assume that the inconsistent findings of the previous studies of other researcher, as described above, may be explained by the fact, that testosterone is not associated with unfairness perception per se, but promotes the fine-tuning of economic decisions in an intergroup situation. In that way testosterone may be rather linked to ingroup favoritism and/or outgroup hostility, the two characteristics of parochialism altruism, as indicated by the converging evidence from our own studies. The above-mentioned studies tested male soccer fans, who maintain long-term rivalries to other teams and show a strong, genuine feeling of group affiliation that might even compare to a 'tribal identity' (*Van Vugt & Park, 2010*). As such, soccer fans represent a natural social group with a strong emotional involvement that is suitable to study parochial altruism (*Weisel & Böhm, 2015*). Here, we investigated whether testosterone similarly affects parochial altruism in artificially created groups (study 1) and in a more natural context of political party supporters during the election year 2017 (study 2). This was based on two reasons: first, we wanted to examine whether the link between testosterone and parochialism is stable enough to be observed in two independent groups and in different social settings, i.e., whether the association found between testosterone in the minimal group study 1 could also be observed in the more realistic social setting of study 2. And second, the group commitment of soccer fans as well as the enmity to other soccer teams can vary with the current position of the admired team in the league and the ongoing team competition can create a stressor, which is often observed in unstable social hierarchies, and could have potentially altered or obscured subtle parochial tendencies in our previous studies. By assessing the present participants in two rather stable group settings, i.e., either in the minimal group setting with a fixed group association (study 1) or during the German election year (study 2), we tried to control for these interfering variables.

In the first study, we conducted the UG experiment with men that were divided into two arbitrary groups according to their behavioral performance in a reaction time task that was completed directly before the UG. In the second study, we questioned male participants about their political orientation using a ranking procedure through which we individually identified the favorite political party as well as the one most distant to the participant. Moreover, since memory for uncooperative group members was found to be enhanced in previous studies (*Bell et al., 2012*; *Bell & Buchner, 2009*; *Bell & Buchner, 2012*; *Hechler, Neyer & Kessler, 2016*; *Howard & Rothbart, 1980*), we assessed cued recall performance in a surprise face memory task (old-new decision) that used the same number of new faces as the ones that were used to represent players in the UG. With this we wanted to assess whether there was indeed a memory advantage for ingroup members who showed schema-incongruent behavior in the UG, i.e., a norm violation through an unfair proposal, as suggested previously (e.g., *Hechler, Neyer & Kessler, 2016*). Moreover, we wanted to explore whether enhanced memory for ingroup norm violators might by associated with testosterone, since our previous neuroimaging study showed that there may be a link between testosterone and insular brain activation in this context (*Reimers, Büchel & Diekhof, 2017*). Based on the results of our previous studies with soccer fans, we hypothesized that subjects with a high endogenous testosterone level would display enhanced parochial altruism in both studies.

## MATERIAL AND METHODS

### Study 1

#### Participants of study 1

Sixty-one healthy male students (mean age $\pm$ SD: 24.95 $\pm$ 4.28 years) participated in this study. The participants were recruited via online advertisement on a campus website and by word of mouth. Only subjects that reported no drug or alcohol abuse, no chronic or psychiatric illness and did not take any form of medication, especially hormones, were included in this study. They were paid a show-up fee of 10€ for participation and were told that they could win even more money (up to 5€) depending on their performance. This study was approved by the local ethics committee (*Aerztekammer Hamburg; Ethical Application Ref: PV3948*). All subjects gave written informed consent prior to participation.

#### Procedure—Minimal group study (study 1)

Data collection took place from 2014 to 2015. Before the test day subjects were given instructions and were handed out five 2 ml polypropylene Eppendorf tubes for the saliva samples, which they had to collect themselves at home the morning of the test day. They had to start with the first sample directly after waking up and then had to collect another four samples with 30 min in between. During the sampling period of 2 h subjects were told to refrain from eating, smoking and drinking anything but water. Directly after the first sample tooth brushing was allowed, however it had to be finished at least 15 min before the second sample, to prevent contamination by micro-bleeding.

Upon arrival at the test facility, the test procedure started with the group assignment, which was a pencil and paper maze task. This task was intended to create an artificial group formation that was unrelated to the experiment itself. Such a minimal group paradigm (MGP) has previously been shown to evoke ingroup favoritism and intergroup bias (*Tajfel et al., 1971*). Subjects went through the maze. After seven seconds time was stopped. Participants were instructed that, according to the distance they covered in the maze, they were assigned to one of the two groups. The two groups were named after two famous cartoon characters, which were used as the group icons in the following experiments. To make this group assignment even more authentic to the subjects, maze templates that showed the cut-off distance dividing the two groups were shown to the subjects. In reality, the group assignment was pre-determined by the experimenters to ensure an equal distribution among the two groups. For subjects who were supposed to be in the 'fast group' another template was used than for subjects who were assigned to the 'slow group'. After the group assignment, subjects were given written instructions that explained the rules of the UG (see *Diekhof, Wittmer & Reimers, 2014*; *Reimers, Büchel & Diekhof, 2017* for comparison).

The UG was designed as a computer-based experiment that was run using the Presentation software by NBS (Neurobehavioral Systems). A short training version was completed and all further questions were answered. Subsequently, the UG was played in the role of the responder (see below), followed by two questions asking about hypothetical offers in the role of the proposer. After that subjects completed the Barratt-Impulsiveness-Scale (BIS; *Patton, Stanford & Barratt, 1995*) and answered several questions from the German
socio-economic panel that measure trust, positive and negative reciprocity (*Dohmen et al., 2008*). Saliva samples were frozen at −20 °C until further analysis.

### Ultimatum Game in a minimal group context (study 1)

One famous and often applied economic decision task is the UG (*Güth, Schmittberger & Schwarze, 1982*). In the UG two players, the proposer and the responder, interact in an economic exchange. The proposer has to make an offer about how to split a fixed sum of money (or experimental points) to the responder. If the responder accepts the offer, both players get paid according to the proposed share. But in case he rejects the offer, both players get nothing. Despite the costs, humans tend to offer almost equal shares and tend to reject unfair offers lower than 20% (*Güth, Schmittberger & Schwarze, 1982*; *Henrich et al., 2005*).

Here we applied a computer-based intergroup version of the UG (intergroup-UG) that was similar to previously applied intergroup-UGs (*Diekhof, Wittmer & Reimers, 2014*; *Reimers, Büchel & Diekhof, 2017*). The participants of the present study thereby acted in the role of the responder and could accept or reject any offers made to them. Yet in contrast to these previous studies, we will focus the analysis on the first session of the intergroup-UG here. In the first session of the intergroup-UG, subjects play for their personal reward and do not receive any further instructions regarding a competition between groups for an extra group bonus, which would be the case in the second session of the intergroup-UG. In our previous studies (see for example in *Diekhof, Wittmer & Reimers, 2014*), the first session was termed the ''neutral context'', since there the group affiliation is not explicitly addressed as being important for the accomplishment of the task goal, i.e., one's individual reward. In contrast, the second session was referred to as the ''competitive context'', since there we also introduced the second task goal, which was to achieve an additional bonus in form of a group reward for behavior that maximizes the outcome of the group. In that way, the ''neutral context'' of the first session of the intergroup-UG should reveal the latent parochial tendency of the person which is not enforced by an explicit instruction that refers to the group identity of the proposers (see also *Reimers, Büchel & Diekhof, 2017*). Moreover, the interpretation of the results from the competitive, second session of the intergroup-UG is problematic. This is because the ''neutral context'' of session 1 always precedes the ''competitive context'' of session 2. Consequently, transfer effects cannot be ruled out from the naïve context of session 1 to the competition in session 2. For example, some subjects might, after short reflection, regret that they rejected any offer during session 1 and could switch to a more selfish-strategy, especially if they consider the group reward rather unlikely to achieve. This could be particularly the case in a study in which ingroup cohesion might be expected to be rather lax (as in study 1) or in which more than two groups are competing for the same group reward (like in study 2 below). In particular, when comparing the results from different studies for consistency this could increase the risk of a study-specific bias in the ''competitive context'' of session 2 that might obscure or contaminate the already subtle effect of testosterone on behavior.

In study 1, the first session consisted of 40 single-shot interactions, during which subjects faced either an ingroup or an outgroup proposer (i.e., 20 trials each). The proposers were

always endowed with 10 points and half of their offers were either fair (i.e., 4 or 5 out of 10 points) or unfair (i.e., 1, 2, or 3 out of 10 points). Offer types and group membership of the proposers were pseudorandomized and counterbalanced for condition transitions. Fair offers contained either 4 points (two trials/group) or 5 points (eight trials/group) out of 10 points, while unfair offers were determined as 1 point (four trials/group), 2 points (four trials/group) or 3 points (two trials/group). This means that 80% of unfair proposals offered 20% or less of the initial endowment in the present study. Since 20% is assumed to be the threshold at which subjects start to reject more and more unfair offers (*Güth, Schmittberger & Schwarze, 1982*; *Henrich et al., 2005*), a rejection rate equal to or exceeding 80% of all unfair offers, i.e., comprising those from both the in- and outgroup in our version of the intergroup-UG, was assumed to reflect a high degree of inequity aversion.

In the intergroup-UG the proposers were introduced with a photo and their first name including initials of the surname to increase authenticity. The photographs showed a frontal view of the face with a neutral expression. The pictures were taken from other male students of Hamburg University about five years before the present study. To control for familiarity, we asked all participants after the experiment whether they knew anyone, but this was not the case. The group membership was indicated by small group icons (i.e., the cartoon characters). To further emphasize the social nature of the task, subjects were told that the proposers were former participants and that participants of the present study would now decide about their payoff. However, all proposers and their offers were pre-determined by the experimenter.

Before starting the intergroup-UG, participants were instructed to decide about each offer according to their individual preference. As they were told in the instructions, each decision determined their additional payment and the accumulated points would be translated into Euros. A total of 2.50 Euros was the maximum reward that could be accumulated in the first session. Subjects also performed the intergroup-UG in a "competitive context" (session 2) for the same reward and the additional chance to win a group bonus (see *Diekhof, Wittmer & Reimers, 2014* for comparison). The data from the second session were however not analyzed here, for reasons outlined above.

After having completed the intergroup-UG (total duration of 30 min), subjects were asked to switch to the role of the proposer and to make two offers to an anonymous ingroup and outgroup member.

### Analysis of hormonal parameters from human saliva (study 1)

After having collected and frozen all samples, the saliva samples were thawed at 26 °C and then vortexed and centrifuged at RCF 604× g in a common Eppendorf Minispin centrifuge for 5 min to discard mucus and other residuals. The five morning samples were pooled into one aliquot by extracting an equal volume of each sample and mixing them together. This was done to control for the pulsatile secretion pattern of testosterone and ensured that the morning peak concentration was captured by the pooled aliquot sample. Samples that looked contaminated (e.g., had no transparent color or contained traces of blood) were discarded. Testosterone concentrations in the aliquot samples were analyzed using two enzyme-linked immunosorbent assay (ELISA) kits by Demeditec Diagnostics (sensitivity

= 2.2 pg/ml). The denoted intra-assay coefficient of variation is indicated as 6.58% at 90.8 pg/ml and the interassay variation is given as 7.4% at 74.3 pg/ml. All aliquots were assayed twice and two control samples, one with a low and another one with a high concentration, were included.

### Statistical analyses of study 1

Mean rejection rates in all experimental conditions, i.e., group membership of proposer (ingroup, outgroup), and type of offer (fair, unfair), were calculated for each subject. The testosterone concentration was standardized ($z$-transformed). The rejection rates were then submitted to a repeated-measures ANOVA, to test whether they were affected by group membership, offer type and standardized testosterone level.

In case of a significant Mauchly-test for sphericity in the ANOVA, results were reported using the Greenhouse-Geisser-corrected values. Post-hoc assessment of the interactions with standardized testosterone were further subjected to simple slope analyses based on the estimated marginal means. We thereby compared the interaction effect in men with high testosterone (standardized testosterone at +1 SD) and those with low testosterone (standardized testosterone at −1 SD) using $t$-tests. P-values smaller than 0.05, two-tailed, were considered as significant.

## Study 2
### Participants of study 2

Thirty-four healthy male students (mean age ± SD: 25.06 ± 4.46 years) participated in the second study. The subjects were recruited via online advertisement on a campus website and by word of mouth. Participants reported no drug or alcohol abuse, and were free of any chronic illness including disorders of the hormone system as well as psychiatric or neurological disorders. They were also free of medication. A participation fee and an additional monetary reward related to the points acquired during the intergroup-UG were paid (see study 1). All subjects gave written informed consent and the study was approved by the local ethics committee (*Aerztekammer Hamburg*).

### Procedure of political supporter study (study 2)

Key to participation in study 2 was an interest in politics, which was assessed by the German Scale for Political Interest (the PIKS questionnaire; *Otto & Bacherle, 2011*). The PIKS questionnaire was sent by e-mail to the potential participant. The five questions on political interest and involvement were to be answered on a 5-point-Likert-scale, possible options ranging from 5, "applies fully", to 1, "doesn't apply at all", resulting in a mean PIKS-score (±SD) of 4.14 (±0.59) for all participants. To determine group affiliation, the participants were additionally asked to rank the six political parties, which had a chance to win seats in the German parliament (*Bundestag*) according to their preference in the upcoming vote. The rating was thereby to be based on the shared values a participant held with the parties. The political parties comprised the four mainstream parties, i.e., the *SPD (Social Democrats)*, the *CDU (Christian Democrats)*, the *Greens*, and the *FDP (Free Liberals)*, as well as two parties from the extreme left and right wing, i.e., *The Left* and the *AfD (Alternative for Germany)*, which all succeeded to acquire seats in the German

*Bundestag* in 2017. The rating procedure was carried out with attention to privacy of the participant in order to avoid social pressure and to rule out bias from the experimenter's side. Based on the political ranking, the participants were ascribed an affiliated ingroup (ranked as 1), while all other parties were considered as outgroups with varying degree of social distance depending on their rank. The party with a ranking of 6 was considered as the most distant outgroup. For subsequent trend analysis, these individual ranks were used. In order to compare the results of study 2 to those of study 1, we averaged the data (UG rejection rates) from ranks 1 to 3 as well as the data from ranks 4 and 6. By this we created one relatively close 'ingroup' (ranks 1 to 3) that shared similar political values from one end of the political spectrum (e.g., the left, more liberal wing) with the respondent, while the 'outgroup' (ranks 4–6) consisted of more distant parties from the other end of the political spectrum (e.g., the more conservative parties oriented towards the right wing or vice versa). Otherwise the procedure of study 2 closely matched the one employed in study 1, including the same questionnaires and saliva sampling procedure. Data collection took place in 2017. The complete experiment lasted about 1 h and 15 min in total, which included the two sessions of the UG (see above) followed by a short distraction period and a face memory task.

### Ultimatum game in a political context (study 2)

The structure of the intergroup-UG strongly resembled the one of the minimal group study (approximate total duration of 30 min), starting with participants in the role of the responder and finally switching to the role of the proposer. Yet, in contrast to study 1, proposals came from six different groups that comprised the supporters from the six political parties. Again, each proposer was indicated by a unique photograph. The photographs were taken from the anonymous experimental picture stock of the department as well as from other universities. The pictures to be used in the game had previously been tested for approachability, making sure that none of the faces evoked a biased response in the participants. The approachability test was performed in an independent sample of 30 students, male and female, who rated each of the faces on a seven-point Likert-scale answering the question: "Would you ask this stranger for directions?" Based on the rating, three faces remained unused in the experiment due to an extraordinary mean approachability rating, the others were equally distributed into the categories of (un)fairness and political inclination by approachability.

The first, unbiased session of the intergroup-UG comprised 48 single-shot computer game interactions with an apparently real, but actually fictional opponent indicated by the unique photograph. In an introductory text, participants were falsely informed about a nation-wide study in which other students had already participated, whose pictures, proposals and party affiliation participants were about to see in the game. From each political party four offers were considered as (rather) fair (with two proposers offering either 4 or 5 points of 10), while the remaining four offers were unfair and included one offer of 1 point and 2 points, each, as well as two offers of 3 points. This resulted in a different threshold for determination of the individual decision strategy in study 2. Here, only 50% of the unfair proposals offered 20% or less of the initial endowment. This is why

a mean rejection rate equal to or exceeding 50% across all unfair offers of the 6 ranks was assumed to reflect a high degree of inequity aversion.

### Cued recall task (memory for faces) of study 2

In the second study we also wanted to test (1) whether participants differed in their overall ability to recollect the identity of faces from previous interactions, in particular unfair ingroup members, (2) whether they showed a differential recollection for persons who treated them unfair or fair on previous trials, and (3) whether recollection ability varied in relation to testosterone level. Similar procedures have already been employed in studies using related paradigms like the dictator game and demonstrated a memory advantage during recollection of the identity of unfair interaction partners, particularly for those from the ingroup (*Hechler, Neyer & Kessler, 2016*). The face memory task followed a short distraction period, in which participants filled out several questionnaires including the Trust and Reciprocity questionnaire (*Dohmen et al., 2008*) and a demographic questionnaire. In addition, they were asked to see themselves as proposer in an UG and make their own proposals to future opponents. These proposals were to be made to 6 hypothetical future respondents, portrayed as grey anonymous silhouettes, each of whom was stated to be favoring one of the aforementioned six political parties. Offers were allowed to range between 1 and 5 points. Another assignment consisted of the same proposal task with a variation: instead of the UG, the participants were playing a dictator game, in which none of the anonymous partners would be able to reject their offer.

The subsequent cued recall task was presented to the participants by surprise. The 48 faces used in the two sessions of the UG were pseudorandomized and mixed with 48 new faces which had been selected from the same database and tested for approachability as stated above. Appearing one after another like in the UG, none of the faces were juxtaposed with any information. Instead, participants were asked to indicate with a button press whether the face was known or unbeknownst to them (old vs. new decision). Each trial ended with the subject's decision. The aim of this face memory task was to assess individual memory for unfair opponents in relationship with the ingroup-outgroup dynamic (*Hechler, Neyer & Kessler, 2016*).

### Hormone analysis (study 2)

The analysis of saliva samples followed the procedure described above (see study 1 above).

### Statistical analyses (study 2)

Data analysis of the UG data in study 2 paralleled the analysis in study 1. We performed a repeated measures ANOVA to assess the influence of group affiliation, offer type and standardized testosterone on rejection rates. Post-hoc testing also followed the procedure in study 1. In order to also assess the six ranks in more detail, we further performed a linear trend analysis to determine the influence of social distance (political party ranking from 1 to 6) on rejection rates, as outlined in detail in the results section. In contrast to study 1, study 2 also incorporated a cued recall task, in which we wanted to examine differences in the ability to recollect previously encountered face identities as a function of their unfairness and social distance as well as in relation to testosterone. For this purpose

we analyzed the hit rates for recalling the unfair proposers from the 'ingroup' (ranks 1–3) and the outgroup (ranks 4–6).

## RESULTS

### Results of study 1

Replicating previous results (*Diekhof, Wittmer & Reimers, 2014*; *Reimers, Büchel & Diekhof, 2017*; *Reimers & Diekhof, 2015*) a clear tendency towards parochialism was found. A 2 ("group": ingroup, outgroup) $\times$ 2 ("offer": unfair, fair) repeated-measures ANOVA with the covariate "standardized testosterone" revealed a significant two-way interaction between "group" and "offer" ($F_{1,59} = 5.44$; $p = 0.023$; partial eta$^2 = 0.08$). In addition, we also found a main effect of "group" ($F_{1,59} = 10.56$; $p = 0.002$; partial eta$^2 = 0.15$) as well as of "offer" ($F_{1,59} = 194.07$; $p < 0.001$; partial eta$^2 = 0.77$) (see also Table 1). Apart from that, there was a significant three-way interaction between "offer", "group" and "standardized testosterone" ($F_{1,59} = 7.82$; $p = 0.007$; partial eta$^2 = 0.12$). Post-hoc simple slope analysis of the estimated marginal means indicated that unfair offers were more likely to be rejected if they came from an outgroup proposer than from an ingroup member, however only if the respondents were men with relatively high testosterone levels ($t_{(59)} = 3.64$, $p = 0.001$). For men with low testosterone the same comparison was not significant ($t_{(59)} = 0.93$, $p = 0.358$) (Bonferroni-corrected threshold: $p = 0.025$). In case of rather fair offers, which also included the 40% offers, both participants with low and high testosterone level tended to be more likely to reject offers from outgroup than from ingroup members, yet these differences were only numerical as they missed the threshold for significance in a two-tailed $t$-test (high testosterone: $t_{(59)} = 1.81$, $p = 0.076$; low testosterone: $t_{(59)} = 1.96$, $p = 0.055$) (Bonferroni-corrected threshold: $p = 0.025$). The results from the simple slope analysis of the estimated marginal means were confirmed by an analysis based on the complete data that compared the subjects with a testosterone concentration above the standardized mean (0) as the high T group ($n = 29$) with those with a testosterone level below 0, i.e., the low T group ($n = 32$). Accordingly, only the high T group showed increased rejections of the unfair outgroup offers compared to unfair ingroup offers ($t_{(28)} = 2.97$, $p = 0.006$), while this was not the case in the low T group ($t_{(31)} = 1.40$, $p = 0.173$). Again both groups showed enhanced rejection rates when rather fair offers were made by an outgroup member, whereby only in the low T group this difference was significant (high T group: $t_{(28)} = 1.84$, $p = 0.077$; low T group: $t_{(31)} = 2.13$, $p = 0.041$) (see Fig. 1A). Yet, the latter effect did not survive the Bonferroni correction ($p < 0.025$), which needed to be applied since the same data were also used in the direct comparisons of rejection rates between the testosterone groups that yielded no significant differences. We only found a non-significant increase in the Delta of rejections of unfair offers made by the outgroup minus those from the ingroup (Delta$_{(high\ T)}$: mean $= 17.93\%$; sem $= 6.05\%$; Delta$_{(low\ T)}$: mean $= 5.31\%$; sem $= 3.81\%$; $t_{(59)} = 1.77$, $p = 0.084$). Collectively, these results indicate a somewhat greater parochialistic tendency in men at the high end of the testosterone distribution, who differentiated more between unfair proposals from the out- and the ingroup.

In a second step, we also looked at the proposals made by the participants to anonymous ingroup and outgroup members after having completed the UG. A repeated measures

**Table 1 Study 1—minimal group design ($n = 61$).** Results of the repeated measures ANOVA of rejection rates (%)

| Main effect or interaction | F-value | df | p-value | Partial eta² |
|---|---|---|---|---|
| Offer[*] | 194.07 | 1, 59 | <0.001 | 0.77 |
| Group[*] | 10.56 | 1, 59 | 0.002 | 0.15 |
| z-testosterone | 0.45 | 1, 59 | 0.505 | 0.01 |
| Offer × group[*] | 5.44 | 1, 59 | 0.023 | 0.08 |
| Offer × z-testosterone | 0.10 | 1, 59 | 0.758 | <0.01 |
| Group × z-testosterone | 1.54 | 1, 59 | 0.220 | 0.03 |
| Offer × group × z-testosterone[*] | 7.82 | 1, 59 | 0.007 | 0.12 |

**Notes.**
[*]Significant effects ($p < 0.05$) are marked with an asterisk.

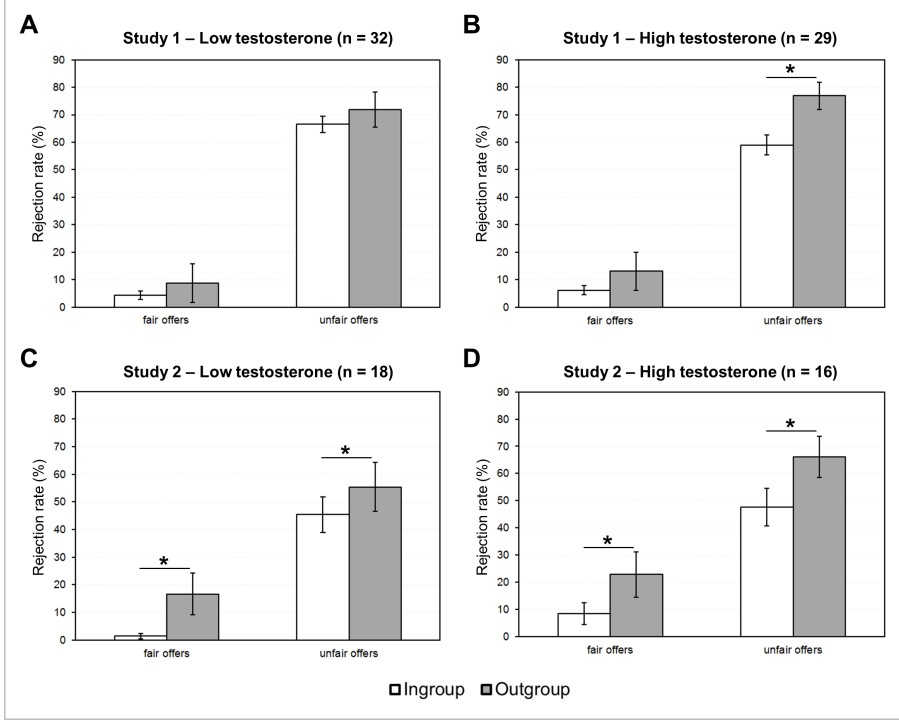

**Figure 1 Rejection rates of ingroup and outgroup offers in association with testosterone.** Data are subdivided by testosterone group and study. The low testosterone group comprised individuals with a standardized testosterone level below the standardized mean (0), whereas the high testosterone group included the subjects with above-average testosterone. The results of study 1 indicate that men with low testosterone (A) were less inclined to reject unfair outgroup offers than men with higher testosterone (B), while the former group showed a parochialism bias only in response to rather fair offers. Study 2 indicates no clear effect of testosterone on the parochialism bias, when subdividing the complete sample in two groups with low (C) versus high (D) testosterone. However the analysis of the estimated marginal means that reflect the most extreme cases at the high versus low spectrum of testosterone, showed a similar parochialism bias in high testosterone men as observed in study 1 (see text). Significant differences (Bonferroni-corrected threshold of $p < 0.025$) in rejection rates are marked with an asterisk.

ANOVA with the factor "group" and the covariate standardized testosterone only showed a significant main effect of "group" ($F_{1,59} = 19.48$; $p < 0.001$; partial eta$^2$ = 0.25), while no significant interaction with nor a main effect of standardized testosterone could be observed. A post hoc $t$-test showed that participants offered significantly more points to an anonymous ingroup member (mean $\pm$ sem = 4.1 $\pm$ 0.1 points) compared to an outgroup member (mean $\pm$ sem = 3.4 $\pm$ 0.2 points) ($t_{(60)} = 4.45$, $p < 0.001$), yet this effect was independent of testosterone.

Finally, being in the 'fast' or 'slow group' did not influence the rejection rates, which was tested by re-running the first ANOVA including the subject's artificial group membership. Also when comparing rejection rates of the 'fast' and 'slow group' directly, no significant differences emerged ($p > 0.347$).

## Results of study 2

Study 2 was intended to examine whether the results related to parochial altruism in study 1 could also be found in a more natural group context (supporters of political parties tested during the German election year 2017). First of all, we found that the political preferences of our student sample were clearly oriented towards the left wing parties (SPD, The Greens, The Left), which was also reflected by the mean ranking across participants (lower mean ranks indicate a more favorable rating (mean $\pm$ sem): SPD = 2.44 $\pm$ 0.23, The Left = 2.94 $\pm$ 0.29, The Greens = 2.97 $\pm$ 0.23, CDU = 3.06 $\pm$ 0.22, FDP = 4.03 $\pm$ 0.23, AfD = 5.56 $\pm$ 0.20). Moreover, in line with study 1 and our previous results (*Diekhof, Wittmer & Reimers, 2014*; *Reimers, Büchel & Diekhof, 2017*; *Reimers & Diekhof, 2015*) we found a significant two-way interaction ($F_{3.69,121.78} = 2.81$; $p = 0.032$; partial eta$^2$ = 0.079) in the repeated-measures ANOVA with the factors rank of "political preference" (6 ranks) and "offer" (unfair, fair). This was also reflected by a significant linear trend with increasing social distance, as determined by an univariate ANOVA with the independent factor "rank" and the dependent variable "mean rejection rate for unfair offers" ($F_{1,203} = 9.94$; $p = 0.002$). When comparing rejection rates between the most extreme cases, i.e., the first rank (one's favorite political party) and the last rank (the 6th rank, which is associated with the least liked party with the most distant political values in the ranking) we found significantly enhanced rejection rates for offers by rank 6, both in the context of fair and unfair offers ($p < 0.001$). In the comparison of the rank 1 with the other ranks, we also documented another two significant differences in the treatment of unfair offers by supporters of distant political parties ranked as 4 and 5, whose rejection was also more frequent ($p < 0.001$) (see Fig. 2, which displays the rejection rates related to unfair offers). All these differences survived a Bonferroni corrected threshold for multiple comparisons of 0.01. This means, that the design of study 2 was also suitable to show parochialistic tendencies amongst the supporters of political parties. Also the subsequent subdivision of ranks into the "ingroup" (ranks 1–3) and the "outgroup" (ranks 4–6) below is in line with these observations of a differential treatment of ranks 4–6, but not ranks 2 and 3 compared to rank 1.

More importantly, the main question was whether these parochialistic tendencies were also related to endogenous testosterone as already documented in study 1. For this purpose
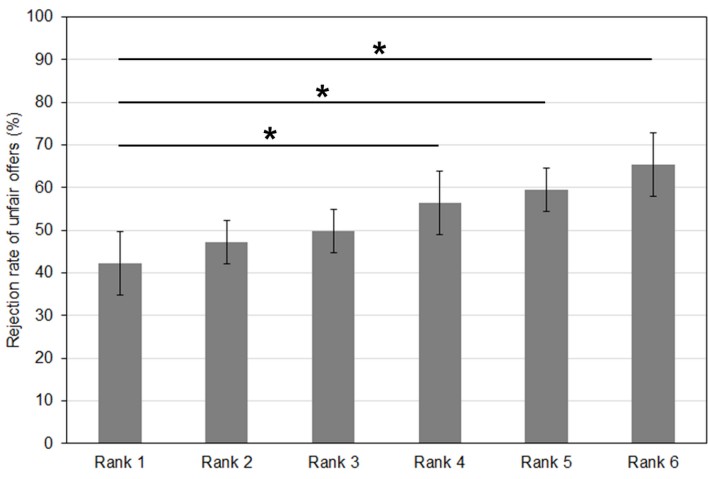

**Figure 2  Rejection rates for unfair offers increase with increasing social distance of political party sup-
porters in study 2.** Significant differences are observed between the first rank, i.e., supporters of one's fa-
vorite political party, and the last three ranks ($p \leq 0.002$) (Bonferroni-corrected threshold of $p < 0.01$).

and to make the data of study 2 (6 groups) most comparable to those of study 1 (2 groups)
the rejection rates from ranks 1 to 3 were combined as the "ingroup", while the remaining
ranks represented the "outgroup". These data were then subjected to a repeated-measures
ANOVA with the factors "offer" and "group" and the covariate standardized testosterone.
In contrast to study 1 this ANOVA only revealed two main effects of "group" and "offer",
while the three-way interaction of the factors merely approached statistical trend level
(see Table 2 for details). Nevertheless, following the procedure of study 1 we subjected
the estimated marginal means to an exploratory simple slope analysis. This exploratory
analysis showed that, similar to study one, only men with very high testosterone showed
a significantly increased rejection rate for unfair offers made by outgroup compared
to ingroup members ($t_{(32)} = 4.46$, $p < 0.001$), while this was not the case in men with
low testosterone ($t_{(32)} = 1.78$, $p = 0.085$) (Bonferroni-corrected threshold: $p < 0.025$).
Further, fair offers of the outgroup were more often rejected than fair ingroup offers
by both testosterone groups (high testosterone: $t_{(32)} = 2.68$, $p = 0.012$; low testosterone:
$t_{(32)} = 2.50$, $p = 0.018$) (Bonferroni-corrected threshold: $p < 0.025$). If subdividing the
sample by the standardized mean of testosterone (0) in a group with above ($n = 16$)
and one below average testosterone ($n = 18$) direct comparisons showed no differential
rejection pattern as originally demonstrated in study 1. Further paired $t$-tests comparing
rejection rates for fair ingroup vs. outgroup offers and unfair ingroup vs. outgroup offers
in the two testosterone groups all yielded significant differences, that were in line with
the interaction effect between "group" and "offer" (all $p < 0.020$; Bonferroni-corrected
threshold: $p < 0.025$; see Fig. 1B). This suggests that there was no clear association between
testosterone and parochialism in this smaller sample of political supporters.

Similar to study 1, the ANOVA of the proposals the participants made to anonymous
members of the first and the last rank after completion of the UG yielded no interaction
with standardized testosterone, yet only revealed a main effect of "group" ($F_{1,32} = 26.40$;
**Table 2 Study 2—political party supporters ($n = 34$).** Results of the repeated measures ANOVA of rejection rates (%).

| Main effect or interaction | F-value | df | p-value | Partial eta$^2$ |
|---|---|---|---|---|
| Offer[*] | 72.26 | 1, 32 | <0.001 | 0.69 |
| Group[*] | 19.83 | 1, 32 | <0.001 | 0.38 |
| z-testosterone | 0.60 | 1, 32 | 0.443 | 0.02 |
| Offer × group | 0.06 | 1, 32 | 0.802 | <0.01 |
| Offer × z-testosterone | 0.22 | 1, 32 | 0.642 | <0.01 |
| Group × z-testosterone | 0.99 | 1, 32 | 0.327 | 0.03 |
| Offer × group × z-testosterone | 2.81 | 1, 32 | 0.104 | 0.08 |

**Notes.**
[*]Significant effects ($p < 0.05$) are marked with an asterisk.

$p < 0.001$; partial eta$^2 = 0.45$). A post hoc $t$-test showed that participants offered significantly more points to a member of the closest rank 1 (mean ± sem = 4.6 ± 0.1 points) compared to a member of most distant rank 6 (mean ± sem = 3.0 ± 0.3 points) ($t_{(33)} = 5.17$, $p < 0.001$).

Finally, in study 2 subjects performed a surprise cued recall task that was similar to the one used by *Hechler, Neyer & Kessler (2016)*. However, in our task version the group identity of the faces shown was not indicated during the memory test, but all faces were shown without any notion of group identity. It has been suggested that there may be a memory advantage for individuals of one's own group, especially if they exhibited schema-incongruent behavior (e.g., behaved unfairly) (*Hechler, Neyer & Kessler, 2016*). We performed a repeated-measures ANOVA with the factors "offer" and "group" and the covariate standardized testosterone to assess their effects on memory performance. This yielded a significant two-way interaction between "offer" and testosterone ($F_{1,32} = 5.57$; $p = 0.025$; partial eta$^2 = 0.15$), but no other significant effect emerged (Table 3). The simple slope analysis of the estimated marginal means revealed one significant difference that was restricted to subjects with low testosterone. These participants showed a better recall rate for fair than unfair proposers ($t_{(32)} = 2.53$, $p = 0.017$), yet independent of group (Bonferroni-corrected threshold: $p < 0.025$). In the analysis of all subjects with a testosterone level below the standardized mean ($n = 18$) an increased hit rate for the identity of fair proposers per se (mean ± sem = 51.3% ± 3.8% correct) compared to unfair proposers could also be observed (mean ± sem = 44.0% ± 4.6% points), yet this numerical difference was not significant ($t_{(17)} = 1.47$, $p = 0.11$). Similarly, the numerically difference in recall of unfair proposers between men with high testosterone ($n = 16$; mean ± sem = 52.6% ± 3.6% correct) and those with low testosterone ($n = 18$; mean ± sem = 44.0% ± 4.6% points) did not reach significance ($t_{(32)} = 1.450$, $p = 0.157$). It thus remains to be determined in a bigger sample whether testosterone may promote memory for unfairness and whether better memory performance indeed depends on group affiliation as previously suggested.

**Table 3 Study 2—political party supporters ($n = 34$).** Results of the repeated measures ANOVA of cued recall hit rate (%).

| Main effect or interaction | $F$-value | df | $p$-value | Partial eta$^2$ |
|---|---|---|---|---|
| Offer | 1.44 | 1, 32 | 0.238 | 0.04 |
| Group | 0.42 | 1, 32 | 0.521 | 0.01 |
| z-testosterone | 0.55 | 1, 32 | 0.464 | 0.02 |
| Offer × group | 0.04 | 1, 32 | 0.846 | <0.01 |
| Offer × z-testosterone[*] | 5.57 | 1, 32 | 0.025 | 0.15 |
| Group × z-testosterone | 2.21 | 1, 32 | 0.147 | 0.06 |
| Offer × group × z-testosterone | 0.19 | 1, 32 | 0.670 | 0.01 |

**Notes.**
[*]Significant effects ($p < 0.05$) are marked with an asterisk.

## Analysis of the combined data from study 1 and 2

The previous results suggested similar effects of testosterone on parochialistic behavior in the two studies. However, the observations made in study 2 were not significant, which was most likely a result of the smaller sample size. We therefore combined the ingroup and outgroup data from the UG in a meta-analysis of all cases ($n = 95$) and re-ran the repeated-measures ANOVA of the rejection rates described previously, this time with the additional between-subjects factor "study". This analysis confirmed the significant three-way interaction of "offer", "group" and standardized testosterone ($F_{1,92} = 10.65$; $p = 0.002$; partial eta$^2 = 0.10$) as well as the two main effects of the factors "offer" ($F_{1,92} = 222.29$; $p < 0.001$; partial eta$^2 = 0.71$) and "group" ($F_{1,92} = 29.46$; $p < 0.001$; partial eta$^2 = 0.24$) (see also Table 4). Further, the simple slope analysis of the estimated marginal means showed that unfair offers were more often rejected if they came from an outgroup proposer than from an ingroup member, however only if the respondents were men with high testosterone ($t_{(93)} = 5.28$, $p < 0.001$). For men with low testosterone the same comparison yielded no significant difference ($t_{(93)} = 1.71$, $p = 0.091$). In case of the rather fair offers, all participants rejected offers from outgroup members more often than those from ingroup members (high testosterone: $t_{(93)} = 3.61$, $p = 0.001$; low testosterone: $t_{(93)} = 3.58$, $p = 0.001$). Apart from these replications, we also found a two-way interaction between "offer" and "study" ($F_{1,92} = 8.00$, $p = 0.006$; partial eta$^2 = 0.08$). Post-hoc comparisons showed that the participants of study 1 rejected unfair offers more often (mean ± sem = 68.6% ± 4.2%) than the participants of study 2 (mean ± sem = 53.4% ± 5.5%) ($t_{(93)} = 2.2$, $p = 0.03$) indicating a higher degree of inequity aversion in the first sample. We therefore also looked into the questionnaire data from the two studies as these might help to explain these differences in inequity aversion (please note that missing questionnaire data of some subjects explain the reduced degrees of freedom below). We found that the participants of study 1 were significantly more impulsive as indicated by a higher BIS score (BIS$_{study 1}$: mean ± sem = 65.7 ± 1.3) compared to the subjects of study 2 (BIS$_{study 2}$: mean ± sem = 60.4 ± 1.4) ($t_{(88)} = 2.60$, $p = 0.011$). The participants of study 1 also showed a significantly enhanced negative reciprocity score in the trust and reciprocity questionnaire (negative reciprocity$_{study 1}$: mean ± sem = 8.65 ± 0.25; negative reciprocity$_{study 2}$: mean ± sem = 5.52 ± 0.29; $t_{(91)} = 7.94$, $p < 0.001$). At the same time, both the trust and the positive reciprocity

**Table 4 Meta-analysis of combined data from study 1 and 2 ($n = 95$)—results of the repeated measures ANOVA of rejection rates (%) with between subjects factor "study".**

| Main effect or interaction | F-value | df | p-value | Partial eta$^2$ |
|---|---|---|---|---|
| Offer[*] | 222.29 | 1,92 | <0.001 | 0.71 |
| Group[*] | 29.46 | 1,92 | <0.001 | 0.24 |
| z-testosterone | 1.05 | 1,92 | 0.309 | 0.01 |
| study | 1.82 | 1,92 | 0.181 | 0.02 |
| Offer × group | 1.46 | 1,92 | 0.23 | 0.02 |
| Offer × z-testosterone | 0.26 | 1,92 | 0.609 | <0.01 |
| Offer × study[*] | 8.00 | 1,92 | 0.006 | 0.08 |
| Group × z-testosterone | 2.54 | 1,92 | 0.114 | 0.03 |
| Offer × group × z-testosterone[*] | 10.65 | 1,92 | 0.002 | 0.10 |
| Offer × group × study | 2.59 | 1,92 | 0.111 | 0.03 |

**Notes.**
*Significant effects ($p < 0.05$) are marked with an asterisk.

score were significantly reduced in study 1 (trust$_{study 1}$: mean $\pm$ sem = 7.20 $\pm$ 0.21; positive reciprocity$_{study 1}$: mean $\pm$ sem = 4.18 $\pm$ 0.13) compared to study 2 (trust$_{study 2}$: mean $\pm$ sem = 8.48 $\pm$ 0.27; positive reciprocity$_{study 2}$: mean $\pm$ sem = 10.58 $\pm$ 0.24) (trust$_{comparison}$: $t_{(91)} = 3.65$, $p < 0.001$; positive reciprocity$_{comparison}$: $t_{(91)} = 23.57$, $p < 0.001$). These results suggest that the participants from the two studies might have shown a general difference in the way to treat other humans in economic interactions, which might explain the increased rejection rate in relation to unfair offers made in study 1. Interestingly, when correlating of the scores from the trust and reciprocity questionnaire with the degree of political interest in subjects of study 2 (PIKS score), we found a negative relationship between negative reciprocity and PIKS score ($r = -0.396$; $p = 0.021$). Unfortunately, the $p$-value did not survive the Bonferroni-correcte threshold of $p = 0.0167$ (the PIKS score was used in three correlations with the trust, negative and positive reciprocity scores). So we can only speculate that people with a higher political interest, who also show a stronger need to get political information to make an informed voting decision, might be more conscientious, which is why they would show less retaliation, yet more positive reciprocity and trust.

In addition to that, we wanted to further ascertain to what extent the actual degree to which an individual shows a 'parochialism bias', i.e., the differential treatment of unfair offers made by the ingroup and the outgroup, may be related to testosterone. For this, we performed a correlation analysis with the data of all cases ($n = 95$). This analysis revealed a small, yet significant positive correlation between standardized testosterone and the delta of the rejection rate of unfair outgroup minus unfair ingroup offers (Delta$_{out-in}$: $r = 0.245$, $p = 0.017$; see Fig. 3).

Finally, in order to further explore this positive correlation we followed the procedure of our previous study (*Reimers, Büchel & Diekhof, 2017*), in which we subdivided the sample by the individual tendency to treat highly unfair proposals. It has previously been assumed that offers of 20% of the original share are assumed to be the threshold at which rejection rates start to increase tremendously, yet at the same time inter-individual variation of rejection rates increases (*Güth, Schmittberger & Schwarze, 1982*; *Henrich et*

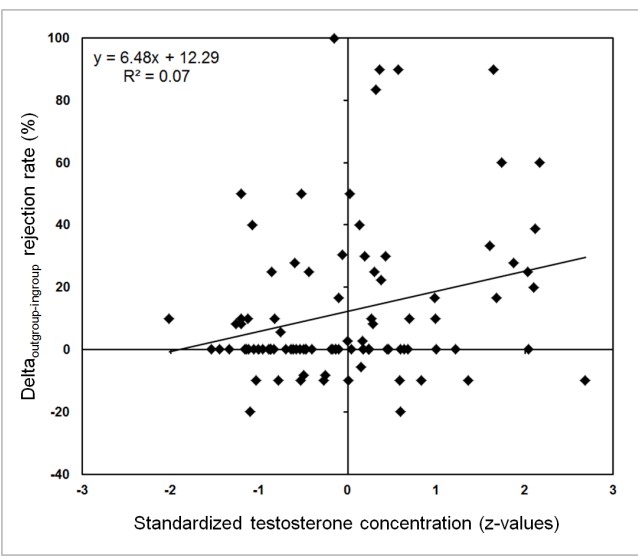

**Figure 3** **Testosterone was positively correlated with the parochialism bias across studies.** Higher endogenous testosterone was significantly associated with higher rejection rates related to unfair outgroup offers relative to unfair ingroup offers ($n = 95$; $r = 0.245$, $p = 0.017$).

*al., 2005*). In the first study 80% of unfair proposals offered 20% or less of the initial endowment, while in study 2 50% of the unfair proposals fell within this range. Thus a rejection rate for unfair offers (regardless of group) that was exceeding these study-specific thresholds of 80% (study 1) and 50% (study 2) was assumed to reflect a high degree of inequity aversion that was marked as the first individual decision strategy (inequity averse subjects). In contrast, rejection rates well below the respective study-specific threshold were considered as reflecting both a high tolerance for unfairness and the motivation to collect as many points as possible for oneself (selfish strategy). Based on this classification, that was exclusively based on the experimental structure of the UG task (i.e., referred to the percentage of highly unfair trials as cut-off value), participants were divided into the two strategy groups (see also *Reimers, Büchel & Diekhof, 2017* for a similar procedure). This sub-division showed that the positive correlation between standardized testosterone and the parochialism bias, i.e., delta of the rejection rate of unfair outgroup minus unfair ingroup offers, stemmed from the selfish subjects ($n = 44$; $r = 0.394$, $p = 0.008$) with their relatively few rejections overall (selfish group: mean rejection rate$_{unfair}$ $\pm$ sem = 34.3% $\pm$ 3.6%), while it was not present in the more inequity averse subjects ($n = 51$; $r = -0.016$, $p = 0.909$). One might argue that the absence of a correlation could be explained by the generally high rejection rates in the latter group (mean rejection rate$_{unfair}$ $\pm$ sem = 88.1% $\pm$ 1.9%) and the resulting ceiling effect. However, it may also be that, similar to *Reimers, Büchel & Diekhof (2017)*, selfish and inequity averse subjects were differently affected by testosterone when deciding how to treat unfair offers made by different groups.

## DISCUSSION

The aim of this research was to test whether an artificial group formation (study 1) and a natural group affiliation (study 2) would both install a sufficient group commitment to measure individual differences in parochial altruism in the intergroup-UG and to investigate how these differences may be related to circulating testosterone. In study 1, the subjects were divided into two artificial groups according to the distance they had covered in a maze task after seven seconds. For study 2, we tested supporters of political parties in the German election year of 2017, who showed a strong interest in politics and had a clear favorite party as well as reservations of varying degree towards the other parties competing in the election. The subjects of both studies acted as responders in a computer-based intergroup-UG with single-shot interactions and offers of varying degrees of unfairness. In the end, participants switched to the role of the proposer and had to decide about hypothetical offers they made to members of their own group or the other group(s). Moreover, study 2 also included a cued recall task to assess memory for previously unfair proposers of the different groups. Altogether, both studies showed a relationship between testosterone and the intergroup bias, thus supporting the assumption that endogenous testosterone may be related to male parochial altruism. However, the correlative nature of the two studies and the relatively small sample sizes limit any strong inferences and render the present results rather preliminary.

### Parochial altruism in the intergroup UG emerged across different studies

As hypothesized and in line with previous studies (*Diekhof, Wittmer & Reimers, 2014*; *Reimers, Büchel & Diekhof, 2017*) a marked pattern of parochial altruism with higher rejection rates in outgroup than ingroup interactions was evident in both studies, which was also reflected by a linear trend for increased social discounting with increasing social distance in study 2 corresponding to the results of earlier studies (e.g., *Strombach et al., 2015*). Such rejections could indicate the willingness to forgo points in order to punish outgroup members, which would reflect a behavior consistent with the theory of parochial altruism (*Choi & Bowles, 2007*). An alternative view would interpret this behavior as spiteful rather than altruistic, with the major aim to minimize the other's payoff in order to equalize relative gain between the interaction partners (*Jensen, 2010*). It has been assumed that spiteful individuals see others as competitors, whose gains negatively affect their own utility (*Espín et al., 2015*). Yet, it is not clear why spitefulness should necessarily follow a parochial pattern or should lead to increased punishment of one social group above the other. We observed that 41 of the 95 participants ($n_{\text{study 1}} = 20$, $n_{\text{study 2}} = 21$) showed a preference to minimize the payoff of unfair outgroup members more than that of unfair ingroup members (Mean Delta$_{\text{out-in}}$ for unfair offers $\pm$ sem = 32.19% $\pm$ 4.19%). This behavior would fit with the theory of parochial altruism (*Choi & Bowles, 2007*). First, these rejections indirectly benefited the relative gain of the ingroup, as did the somewhat lowered fairness norm in ingroup interactions. Second, it also clearly harmed the payoff of the outgroup, yet this happened at the responder's expense, since in the latter case the responder declined a share he could have otherwise acquired for himself. In contrast,

another 14 subjects ($n_{study\ 1} = 11$, $n_{study\ 2} = 3$) preferentially rejected unfair ingroup offers compared to unfair outgroup offers (Mean Delta$_{out-in}$ for unfair offers ± sem = −10.87% ± 1.09%). This decision behavior would rather fit with the theory of indirect reciprocity (*Gintis et al., 2003*; *Yamagishi & Mifune, 2008*), according to which rejections of unfair offers by ingroup members even in single-shot interactions promote the adherence to fairness norms and cooperation within one's group. Altruistic punishment of ingroup unfairness is thereby assumed to be the glue of large anonymous societies that follow the principle of generalized exchange. In addition, harsher punishment of norm violations in the ingroup may not only increase the individual probability to get fairer shares in the future, but could benefit one's social reputation and thus enhance individual social status (*Fehr & Gächter, 2002*; *Yamagishi et al., 2012*). Finally, the remaining 40 subjects ($n_{study\ 1} = 30$, $n_{study\ 2} = 10$) showed no difference in responding to unfair offers from the out- and the ingroup. This latter group also showed significantly higher rejection rates in response to unfair offers per se (mean ± sem = 73.01% ± 5.88%) than the subjects with a negative Delta$_{out-in}$ (mean ± sem = 53.37% ± 4.00%) ($t_{(79)} = 2.78$, $p = 0.007$), which might have indeed reflected a motive driven by increased spitefulness (*Jensen, 2010*). Altogether, the present data do not conform to the assumption that the spitefulness motive was the only driving force of rejections in case of unfair offers, yet it could have motivated a subgroup of subjects, which needs to be determined by future studies.

When looking at the findings of study 1 we found that the minimal group formation task elicited parochial altruism in the same manner as did previous studies with natural groups (e.g., ethnic groups as in *Fershtman & Gneezy, 2001*, or the sample of political supporters tested in study 2). The minimal group formation task resembled previously applied methods of assigning subjects to different groups according to their performance on a meaningless task (like in a task that required the estimation of the number of presented dots). Given previous evidence indicating that even such minimal conditions for group assignment promote an intergroup bias (*Brewer, 1979*; *Tajfel et al., 1971*; *Volz, Kessler & Cramon, 2009*), we expected to find a similar pattern. But note that alternative methods to create group identity have also revealed conflicting findings indicating that norm violations committed by ingroup members are punished more often (e.g., *McLeish & Oxoby, 2007*; *Mendoza, Lane & Amodio, 2014*). Although costly punishment has been proposed to sustain group cooperation (e.g., *Fehr & Gächter, 2002*) and may thus be expected to occur more often in response to unfair ingroup members, we found previously that our version of the intergroup-UG particularly provoked increased punishment of unfair outgroup members by young healthy men, even when our participants played for themselves without a direct intergroup competition (*Diekhof, Wittmer & Reimers, 2014*; *Reimers, Büchel & Diekhof, 2017*).

Apart from similarities study 1 and study 2 nevertheless also exhibited a difference, namely in the percentage of rejections of unfair proposals per se (study 1 > study 2). This might have been related to differences in aspects of the personality of subjects, that were determined by two self-report questionnaires. For one thing, the participants of study 1 showed a higher degree of negative reciprocity, which comprised the increased tendency for retaliation or to harm someone who has previously harmed oneself, while the tendency

to return a favor (positive reciprocity) and the trust in strangers was reduced in comparison of subjects from study 2. For another, the participants of study 1 were also more impulsive. Impatience as an aspect of impulsiveness has previously been shown to influence decisions in the UG. In particular, spitefulness in the UG may be driven by an increased impatience, which would result in increased rejections of unfair offers as well as reduced proposals in general (*Espín et al., 2015*). Interestingly, of the 40 subjects, who showed no difference in responding to unfair offers from the out- and the ingroup 30 were from study 1, which is approximately 50 percent of the sample of study 1 (see above). This would fit with the observed average personality profile of increased negative reciprocity in combination with heightened impulsivity of the men from study 1, and points in the direction of increased spitefulness as one motive for the exceedingly high rejection rates.

## Testosterone may modulate parochial altruism in different social contexts

The major aim of this research project was to assess the relation between endogenous testosterone and parochial altruism in two related, yet distinct social settings. Taken together, we found only weak, yet consistent evidence for a positive connection between testosterone and the preferential punishment of unfairness in outgroup members in the two studies (see Fig. 3). Not surprisingly this correlation was most visible in subjects who exhibited a more selfish decision strategy (i.e., who did not show a general distaste for unfairness and thus overall high rejection rate, but in contrast exhibited a rather flexible rejection style that also differentiated more between the in- and the outgroup). This might indicate a modulatory role of testosterone on the behavioral expression of male parochial altruism as suggested previously (see also *Reimers, Büchel & Diekhof, 2017*). The effects of testosterone on altruistic punishment in the UG have often been subtle or were even contradictory at times (see 'Introduction'). This currently precludes an unequivocal interpretation of the role of testosterone in altruistic punishment in the UG and its interaction with parochialism. For this reason, the present data need to be replicated in other intergroup contexts (e.g., in the comparison of different ethnic groups or in members of different universities). Further, the causal relationship between testosterone and parochial altruism has to be determined by pharmacological intervention studies that ideally test both men and women.

As it stands, the two employed intergroup paradigms are most comparable to the ones used by *Diekhof, Wittmer & Reimers (2014)* and by *Reimers, Büchel & Diekhof (2017)*, who also assessed healthy young men. However, the present results do not correspond to our previous findings. In the behavioral study of *Diekhof, Wittmer & Reimers (2014)* the positive relationship between testosterone and parochial altruism was not observed in the unbiased context, but only occurred when soccer fans transitioned from the neutral session to the competitive part of the intergroup-UG, during which groups explicitly competed for an additional group bonus. This was reflected by the relative enhancement of rejections of rather fair outgroup offers (4:6), which was stronger in soccer fans with a high testosterone level. Moreover, the same subjects showed an increased parochialism bias, but only during the competition, whereas there was no such relationship in the

unbiased session of the intergroup-UG. Finally, when being in the role of a proposer, soccer fans with high testosterone made more generous proposals to members of their ingroup, which could not be observed in the present studies. However, our previous and present behavioral studies differ in some important aspects. While in the previous study (*Diekhof, Wittmer & Reimers, 2014*) participants faced proposers of four different sports teams (i.e., the soccer ingroup, a neutral soccer outgroup, an unknown cricket outgroup and a disliked soccer outgroup), three of whom were directly competing in the German soccer league (*Bundesliga*), the present research either used artificially created group identities or employed a natural group of German voters with high political interest. We can only assume that the differences in the groups under research might have led to different degrees of group commitment, as demonstrated previously by *Weisel & Böhm (2015)*. Despite the use of natural groups in both study 2 and the study by *Diekhof, Wittmer & Reimers (2014)*, who were competing for desired resources, i.e., either for seats in the German *Bundestag* or a high *Bundesliga* ranking, we speculate that soccer fandom evokes a much stronger emotional group affiliation than being a supporter of a certain political party prior to an important election in Germany, even though this was not explicitly tested here. Compared to these natural social settings, even less emotional commitment should be assumed in a member of a minimal group that was based on the performance in a simple reaction time task like in study 1, although this assumption is again speculative since it was not tested here. As already discussed in detail by *Weisel & Böhm (2015)*, the election campaigns of the different mainstream parties in Germany (*SPD, Greens, CDU* and *Free Liberals*) are less emotional than for example in the United States (US). As *Weisel & Böhm (2015)* put it, the political discourse in Germany is mild and the mainstream parties are not as polarized as for example the Democrats and Republicans in the US. Also, many voters may consider them to have more in common than in separation. Only parties on the extreme right or left wing may be considered as distinct, which was also demonstrated here by the ranking of the extreme right wing party, the *AfD*. Of the 34 participants of study 2, 28 categorized the extreme right as rank 6, and another three did so as rank 5. In contrast to interactions between political voters in real life, the context of soccer fandom is characterized by a high degree of enmity between teams and normally the affiliation with one's own team is very strong and emotional, that it may often resemble a tribal identity (*Van Vugt & Park, 2010*). Our previous study only tested subjects who strongly agreed with statements like ''*soccer is my life*'' and who owned not only season tickets for matches of their favorite team, but also went to away matches and owned fan merchandise like bedclothes with a team logo (*Diekhof, Wittmer & Reimers, 2014*). Further, a soccer season comprises 34 weeks of a year with games every weekend, while an election for the German *Bundestag* happens only every 4 years and the hot phase of the election campaign comprises only a handful of weeks directly before the election. Soccer fandom thus requires a constant engagement with the success of one's favorite team as well as real life interactions with other supporters of one's team as well as those from rival teams (criteria that were all fulfilled by the participants of the study performed by *Diekhof, Wittmer & Reimers, 2014*). *Weisel & Böhm (2015)* found less outgroup hate in supporters of political parties than in soccer fans in an economic exchange task, which was also sensitive for the different aspects of parochialism. When
comparing the parochialism bias in relation to unfair offers (i.e., the Delta of rejections of unfair offers made by the outgroup minus those from the ingroup) of the present two and our previous behavioral soccer study (*Diekhof, Wittmer & Reimers, 2014*), we see that the minimal group study 1 had the lowest value (mean ± standard deviation = 11.3% ± 27.8%), which was lower than the bias found in study 2 (mean ± standard deviation = 14.1% ± 19.1%). In contrast, the parochialism bias documented by *Diekhof, Wittmer & Reimers (2014)* when comparing the ingroup and the antagonistic outgroup was much higher (mean ± standard deviation = 24.8% ± 36.5%) as was the bias observed in the neuroimaging study by *Reimers, Büchel & Diekhof (2017)* (mean ± standard deviation = 21.7% ± 29.7%). Further, even the Delta based on the average of the three outgroups of the behavioral soccer study (i.e., the antagonistic soccer team, the neutral soccer team, and the unknown cricket team) minus the ingroup rejection rates remained the highest compared to the present two studies (mean ± standard deviation = 17.6% ± 28.3%). Based on this and the observations made by *Weisel & Böhm (2015)*, we speculate that the emotional engagement might have been highest in the hardcore soccer fans, which could be the reason why we had been unable to observe the rather subtle effect of testosterone on behavior during the unbiased context of the UG there, during which soccer fans already exhibited a very high degree of parochialism per se. Nevertheless, the present result emerged across the two different social settings of study 1 and 2, which may point to a valid, but small effect of testosterone on latent behavioral parochialism during socio-economic interactions outside of the context of soccer fandom.

Finally, we also performed a cued memory recall task in study 2. We did not find a memory advantage for schema-incongruent information in the given social setting (e.g., for unfair proposals made by ingroup members) as suggested previously (*Bell & Buchner, 2012*; *Hechler, Neyer & Kessler, 2016*). However, in contrast to these prior studies group association was not reinstated in the cued recall task, but facial identities were shown without any reference to the group they belonged to. This may explain why the general recall performance ranged around 50% chance level for fair and unfair proposers across ranks (mean ± sem: hit rate for all fair proposers = 51.3% ± 2.7%; hit rate for all unfair proposers = 48.0% ± 3.0%), while the hit rate for new pictures was the highest (mean ± sem: hit rate for new pictures = 57.5% ± 2.0%). We also found a small effect of reduced testosterone on hit rates for fair as compared to unfair proposers (better recall of fair proposers) in the analysis of the estimated marginal means. A recent neuropsychological study with a focus on cue-induced forgetting and recall in relation to salivary testosterone found something similar, namely that low testosterone levels were associated with improved binding of the newly encoded memories to their context cue (*Sterzer et al., 2015*). However, since the present effect could not be ascertained in study 1, in which no cued recall test was performed, and did not occur when the complete sample was subdivided by the mean of standardized testosterone (below or above average testosterone), we cannot infer that this was not just a sporadic finding. Future studies have to more carefully assess the interaction of social distance by fairness in larger samples using different cued recall tasks, with or without indicators of group identity.

## Limitations and future perspective

The present studies have several limitations that need to be addressed by future studies. First, sample size, especially that of study 2, was limited and therefore the power to identify the potentially subtle associations between testosterone and behavior was restricted. Nevertheless, we found similar effects in both studies that also held when combining both data sets.

Second, the present studies identified a single hormone as a correlate of parochial altruism in the intergroup-UG. However, testosterone may not be the only hormone involved in ingroup favoritism and outgroup hostility. There is recent evidence for other hormonal systems to play a role in shaping parochial altruism. For instance, the neuropeptide oxytocin has been shown to drive parochial altruism in men during economic interactions (*De Dreu et al., 2010*). The steroid hormones testosterone and also estrogen have been shown to modulate the expression of the neuropeptides oxytocin and vasopressin (*Liening & Josephs, 2010*; *Soares et al., 2010*), which are both involved in a variety of social behaviors and economic decision making (*Bos et al., 2012*). Further, another steroid hormone that might potentially influence testosterone's effects on social behavior is cortisol. A growing number of studies provide evidence for the *dual-hormone hypothesis*, which states that the effects of testosterone on status-related behavior, such as dominance, depend on the levels of cortisol (*Mehta & Josephs, 2010*). In fact, a recent study has provided initial evidence for the dual-hormone hypothesis in the context of an UG showing that a rise in testosterone was associated with increased acceptance rates of unfair offers in individuals with decreased cortisol levels (*Mehta et al., 2015*). Yet, the dual-hormone hypothesis has not remained undebated and two recent meta-analyses demonstrate only weak effects (*Dekkers et al., 2019*; *Grebe et al., 2019*). These results call for much larger samples for hypothesis testing as well as pre-registered study protocols in the future.

Third, another aspect that currently remains unknown, as it was not in the focus of the present study, is the impact of genetic predisposition on individual differences in parochial altruism. For instance, a genetic polymorphism in the androgen receptor gene, the CAG tandem repeat length, is associated with the sensitivity for circulating androgens such as testosterone (*Chamberlain, Driver & Miesfeld, 1994*). Subjects with a more selfish decision strategy were reported to show a tendency towards shorter repeat lengths and thus supposedly increased androgen sensitivity (*Reimers, Büchel & Diekhof, 2017*), which would help to explain the present findings of a positive correlation between the parochialism bias and testosterone that was particularly evident in men who showed reduced inequity aversion across groups. Other studies also demonstrated nuanced relationships between personality, repeat length, testosterone and aggression (*Geniole et al., 2019*), suggesting that genetic predisposition may be an important moderator of the relation between fluctuating hormones and behavior as well as brain physiology.

Fourth, the present studies as well as our previous ones (e.g., *Diekhof, Wittmer & Reimers, 2014*) were restricted to men. It thus remains to be determined how women would react to intergroup manipulations in the UG and whether endogenous testosterone may be related to female parochial altruism. Evolutionary theories of parochial altruism like the

one formulated by *Choi & Bowles (2007)* or the 'male warrior hypothesis' (*Van Vugt & Park, 2010*) do not make explicit claims about female behavior during group competition or conflict. However, the 'steroid/peptide theory of social bonds' (*Van Anders, Goldey & Kuo, 2011*) makes some suggestions regarding the role of testosterone in the achievement of social goals that may also apply for females. Accordingly, in females testosterone may trigger defensive aggression in a special case of social threat, namely when there is a need to protect her offspring against outside threat. One may therefore speculate that women may also show parochialistic behaviors that are related to testosterone, however these might become particularly evident in situations that involve the core ingroup (e.g., family, mother-infant interaction) rather than an abstract group (like in study 1) or a political view (like in study 2). So it remains to be determined by future studies to what extent female testosterone mediates parochialism in general and parochial altruism in particular.

Finally, the expression of parochial altruism may not only be driven by physiological factors, but may further be shaped and could even be intensified by cultural ramifications like gender or racial stereotypes, socially preferred or sanctioned behaviors and long-standing rivalries between groups. Yet, by thoroughly assessing the physiological basis of parochial altruism we might be better able to formulate hypotheses that address the potential interaction between physiological and cultural factors that may collectively lead to prevailing intergroup biases and may fuel racism across the globe.

## CONCLUSION

In conclusion, the present findings show that high levels of testosterone are linked to behavioral patterns of parochial altruism depending on individual decision strategy. Extending previous data that demonstrated an association between testosterone and parochial altruism in soccer fans (*Diekhof, Wittmer & Reimers, 2014*; *Reimers, Büchel & Diekhof, 2017*; *Reimers & Diekhof, 2015*), the present study revealed a comparable relationship in artificially created groups and supporters of political parties. Based on the concordant findings of studies 1 and 2, it may be assumed that the effect of testosterone on parochial altruism represents an evolutionary conserved neurobiological mechanism that is also detectable in minimal group contexts as well as natural social settings outside the context of soccer fandom. In sum, our results add further evidence to the modulatory role of testosterone in shaping parochial altruism and point to potential future avenues for research aiming to understand the neuroendocrinology underlying this prevalent human behavior.

## ACKNOWLEDGEMENTS

We would like to thank A Kroll for the analysis of hormonal parameters and M Langbehn for helping with programming the computer-based UG and the analysis batches. In addition, we would like to thank all the anonymous participants of this research project.

### Funding
The authors received no funding for this work.

### Competing Interests
The authors declare there are no competing interests.

### Author Contributions
- Luise Reimers conceived and designed the experiments, analyzed the data, prepared figures and/or tables, authored or reviewed drafts of the paper, approved the final draft.
- Eli Kappo, Lucas Stadler and Mostafa Yaqubi performed the experiments, analyzed the data, authored or reviewed drafts of the paper, approved the final draft.
- Esther K. Diekhof conceived and designed the experiments, analyzed the data, contributed reagents/materials/analysis tools, prepared figures and/or tables, authored or reviewed drafts of the paper, approved the final draft.

### Human Ethics
The following information was supplied relating to ethical approvals (i.e., approving body and any reference numbers):

This study was approved by the local ethics committee (Aerztekammer Hamburg; Ethical Application Ref: PV3948).

### Data Availability
The raw data are available in the Supplemental Files.

### Supplemental Information
Supplemental information for this article can be found online at http://dx.doi.org/10.7717/peerj.7537#supplemental-information.

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
