# Peer review of "Endogenous testosterone correlates with parochial altruism in relation to costly punishment in different social settings"

_PeerJ, doi:10.7717/peerj.7537_

## Round 0.1 · original submission · Major Revisions

Thank you very much for your submission to PeerJ. I have been fortunate to receive reviews of your paper from three experts in the field. While all three reviewers noted the merit of your study they all had a number of concerns with its current presentation - particularly with regard to the analyses that you performed and the presentation of your results. Additionally, the reviewers all noted that given your small sample size, especially relative to the number of conditions within which you analyzed it, the power of your study is limited and that you should at the very least acknowledge this limitation.

All three reviewers have provided thoughtful and detailed reviews that will help you improve your article. I encourage you to provide detailed responses to their comments as you revise you article describing what edits you have made and why. Please note, reviewer 2 provide their comments as an attached document that you can download using a link provided below.

Reviewer 1 ·

Basic reporting

The manuscript should be proof-read by a native English speaker. There was awkwardness and grammatical issues throughout. Furthermore, the paper did not follow a clear style, particularly in regards to numbers and use of parentheses.

I also find the terminology throughout the paper confusing. Altruism seems to be used interchangeably with favoritism. This does not seem accurate. It also seems like altruism is being used when spite should be used. In terms of the behavioral ecology literature, altruism is doing something beneficial for someone else (+) at some cost (-) to yourself. Spite is doing something detrimental for someone else (-) at a cost (-) to yourself. This convention seems not to be followed in this manuscript. This makes both the intro and discussion sections confusing. See for example: Bshary R.& Bergmüller R.. 2008. Distinguishing four fundamental approaches to the evolution of helping. J. Evol. Biol. 21, 405–420 (Box 1.1).

Experimental design

Generally, the experimental design was appropriate. However, the writing of it, at times was confusing.

Validity of the findings

The validity of the finding were challenging to interpret, as the results of the ANOVAs were not presented in a typical way. The authors, at times, focused on main effects when there was a significant interaction and did not report or explain the simple main effects of that interaction. I am also a bit unconvinced about findings that did not reach statistical significant and had a small effect size. P-values are not everything, but if it is not significant and there is a small effect size, this is rather unconvincing.

Additional comments

Line 69: What do you mean by unique scale? That is unclear. If you are trying to say this is uniquely human, I disagree. If you mean to a large extent, I do agree.

Line 72 – you go straight into the behavioral econ literature, but I am sure this has been shown in other contexts. Perhaps this would be worth mentioning.

81 – I would remove “than their own”

Line 83-84 – This is what I was talking about in my above comment. Perhaps giving an example would make this more impactful.

86 – I think you need to define exactly what you mean by altruism. Do you mean behaving in a way to benefit someone else at a cost to yourself? Or do you mean ingroup favoritism as in line 71? I don’t think you can use altruism and favoritism interchangeably. I suspect what you really mean here is favoritism. I think bringing in altruism might confuse the issue.

108-109: Clarify - “testosterone has recently been proposed to play an important role in mediating the effect of testosterone”

125-130 – This is a very dense sentence. I would consider making this multiple sentences.

137 – missing word: “even in the absence…”

147 – Given world politics at the moment, I think you need a better justification about why political parties would not have a strong tribal identity. It seems like your explanation in lines 151-155 could also potentially apply to political parties.

160 – please clarify why you did this. Was it to make arbitrary groups? How was performance divided? How explain this in the methods. I might avoid talking about the maze until then so it is more clear.

217 – use of parentheses

238 – This is a bit unclear. Are you using the data from your previous study?

250 - use of parentheses

251-252 – this would be more clear without the use of a double negative. Is problematic rather than is not unproblematic.

255 – I am not sure of your use of the word punishment. This typically mean a consequence of behavior. Here, would they be punishing themselves by being generous? I am not following this.


263 – So you are looking at respondent reactions rather than proposer offers. This is why I was confused in my above comment. You need to say that particiants are the respondent. I assumed they were the proposer.

277 – People often think of the potential motivations for the UG as either inequity averse or selfish. However, selfish behavior could also be a rationally maximizing strategy. As you can’t disentangle these potential strategies with the UG, I don’t like the term pro-selfs. At the very least, you should include an acknowledgement that this might not be selfish, but rather rational. In other words, people acting rationally, may not be responding to inequity at all. This is in contrast to your statement that they have a high tolerance for unfairness. I would prefer you call them self-oriented subjects as you sometimes do in the discussion and figures.

284 – Were novel pictures used in both sessions? In other words, in session 2 did people see some of the same faces they saw in session 1? Would this effect the results?

296 – I am a little confused about why you talk about session 2 at all if you are not going to analyze it.

Line 329 – This actually suggests you are correct and they are being selfish. If rational, they wouldn’t reject offers differently between in-group and out-group. However, when you are setting up the study, the logic from my comment at line 277 maybe should be addressed.

Line 328 or so – You say you ran an ANOVA, but you are not reporting the ANOVA results. This should be reported. Actually, I now think that you are sort of mixing the methods and results sections. I would not report what the pro-selfs did until the results section. It is a bit strange to report some, but not all, statistics here.

348 – Why use nonparametric statistics for post hoc comparisons with an ANOVA?

396 – Was approachability also done in study 1? If no, why not?

400 – In what way was it extraordinary? Do you mean an outlier?

406 and other places – please follow the style of PeerJ. Number 1-9 should be spelled out unless used with units. https://peerj.com/about/author-instructions/

431 – Were the total points available still 10? If so, why did you not let them make generous offers?

Section 2.2.6 – Same comment as the previous statistics section. It feels as if you are mixing results in here. I would prefer you to talk about all of study 1, then all of study 2.

Section 3.1. I feel the way you are presenting the ANOVA is confusing. You should address interaction effects before main effects, as some main effects may not be interpretable depending on the interaction. You need to explain the significant interactions.

479 – You identify a trend here, but this is not significant and the effects size is weak. Therefore, this is not convincing to me as a trend that should be reported.

480 – You seem to be talking about the three-way interaction, but only of team and offer, not at each level of testosterone.

482 – Do you mean there were no significant simple main effects? A post-hoc is typically only for one independent variable whereas simple main effects look at the how the level of each IV is effected at each level of the other IVs.

485 – The test statistic, Z, should be reported as well.

486 – Why do you include the p value in line 485 but not 486? Be consistent.

Section 3.1 – I think a graph of the significant interaction would help the reader interpret your results.

503 – Again, this is not significant and only has a small effect size. Therefore, I am not sure you should interpret this. A graph would help.

522 – Supposedly? If this is not your interpretation then this should be cited. Otherwise I am not clear why you said that.

543 – Please interpret the interaction. A graph would be beneficial.

Section 3.2 – It would help the reader to include means and SEM when you talk about where the differences are in the ANOVA.

569 – I am not following the logic of why the correlation coefficient would decrease and that would show a similar strength…

599 – use of parentheses

607 – I know that p-values are not the only way to evaluate effects, but when you have a trend, to really be convincing, the effect size should be large. Again, this is a small effect size.

599 – This is the first time you have mentioned social discounting.

701-702 – forgoing points to punish others would be spite, not altruism

763 – I again don’t understand why you are calling this altruism when it seems to be spite. Spite seems to me to be a more parsimonious explanation.

766 – Do you mean to begrudge the other player?

Reviewer 2 ·

Basic reporting

no comment

Experimental design

underpowered. The authors have a 2 (offer) x 2 (group) x 2 (decision strategy) x 2 (testosterone - median split) design for study 1, and a 2 (offer) x 6 (rank) x 2 (decision strategy) x 2 (testosterone - median split) for Study 2. Study 1 had n = 61 men, while study 2 had n = 34 men. See my comments in the attached.

Validity of the findings

Nice that the findings from Study 2 are similar (though weaker and more variable) than those of Study 1

Additional comments

please see attachment for more specific comments.

Annotated reviews are not available for download in order to protect the identity of reviewers who chose to remain anonymous.

Reviewer 3 ·

Basic reporting

The authors present a well-written paper on the role of endogenous testosterone levels, intergroup membership, and individual differences attitudes toward inequity in predicting rejection behaviors in the ultimatum game. The authors seemingly replicate their findings across two intergroup settings- one using a minimal group paradigm and another using a naturalistic, political context.

Below, I state some broad comments for the authors to improve their paper:

The authors highlight inconsistencies in the testosterone literature concerning ultimatum game findings. However, they do very little to integrate their study both in the introduction and discussion within the context of those inconsistencies. The authors find that both inter group context (in-group vs. out-group) and the individual difference of decision-making strategies toward inequity serve as possible moderators to explain the inconsistent testosterone-rejection behavior findings. The authors could consider using that framing to integrate their research with the extant literature.

There is little theoretical rationale provided for examining the role of fairness-related decision-making strategies in testing their hypothesis.

Experimental design

There is little information on why the cued memory recall paradigm was included in the study, until I reach the results section of Study 2. What was the theoretical rationale? Was this measure primarily used as a manipulation check? Also, why did the authors not consider looking at the testosterone x group interaction for this particular DV?

What was the rationale for the authors measuring testosterone in the morning and not in the afternoon (testosterone usually shows stable hormone concentrations between noon and 5 pm)? Also, did participants self-administer these hormone samples? In which case, how were the samples stored and transported?

Validity of the findings

The authors perform parametric inferential statistical analyses but follow-up those overarching tests with non-parametric analyses. Is there a rationale for doing so? If there is, the authors may consider including that in their paper. The authors collected nested data, and multilevel modeling may afford them more degrees of freedom in examining such data. If anyone on the author-group possesses expertise to conduct those analyses, I would highly encourage them to do so.

Minor comment: In Study 1- the authors report fairly small degrees of freedom for their error term in spite of having 61 participants and a repeated measures design. I may be missing information that was reported previously that explain these small degrees of freedom and/or the authors conducted analyses within a sample sub-group which may be another alternative explanation for the smaller degrees of freedom. It will be nice to know why the testosterone x team x offer interaction had such small degrees of freedom.

To test the relationship between testosterone and parochial altruism, the authors conducted post-hoc analyses among proself individuals by assessing non-parametric correlations between testosterone levels and unfair offer rejection (ingroup vs outgroup difference scores). These analyses are exploratory (in my view) but extremely interesting. However, I would appreciate if the authors first reported the overall interaction between testosterone x decision-making strategy type predicting rejection rates before delving into simple slope analyses where they could examine correlations between testosterone and rejections across the individuals with different strategies. Further, for the sake of transparency, I also encourage the author to report the 4-way interaction between testosterone x offer x group x strategy in their main analyses.

It is unclear to me if the authors did indeed find an interaction between testosterone x group x offer in Study 2? They do not clearly report whether this interaction was present before they proceed to examine the correlations between testosterone and rejection rates. It would be ideal if the authors followed the same/similar reporting order and style as Study 1 for ease of comprehension for the reader.

My biggest concern with this paper to do with the sample sizes used by the authors to test fairly complex and a large number of statistical analyses (although they do correct for their p values during post-hoc analyses). To demonstrate their effects, the authors also adopt a split-by-condition approach, which in turn reduces their statistical power substantially. However, the replication of the findings broadly across both study 1 and study 2 gives me some confidence in their findings. I encourage the authors to acknowledge sample size as a limitation in their discussion section. I do think that examining the boundary conditions of testosterone’s relationship with parochial altruism is a valuable contribution to the psychoneuroendocrinology literature and hope that the authors will consider conceptually replicating these findings in larger samples in the future.

Further, in order to test the replication of the effects reported across both studies, the authors may consider conducting a mini-meta analysis within their paper. Or figure out a way to combine both datasets and examine the combined effects of the testosterone x group x offer interactions and correlations between testosterone and rejection rates across proself vs. inequity averse groups.

In general, I enjoyed reading the discussion section of the paper, especially the portions that integrated their current work (submitted to this journal) with their prior research (which I am familiar with). However, because the study only comprised of male participants, I would encourage the authors to speculate about how their findings may extend in a mixed sex or all female sample.

---

## Round 0.2 · Minor Revisions

Thank you very much for submitting your revised article for consideration. Two of the reviewers who reviewed your original submission have now provided feedback on your revised article. Both commended you for making substantial changes and improvements to your article. One of the two reviewers still has a few outstanding issues that they raise in their review. If you can address these, I think it will be likely that I will be able to accept your article for publication in PeerJ.

Reviewer 1 ·

Basic reporting

The authors have greatly improved the manuscript from the first draft in terms of presenting the theory, using consistent and clear terms and the presentation of results. The ANOVA tables were particularly helpful. With these changes, this now meets the standards of PeerJ. There were still a few typos and I mention these in the general comments section below.

Experimental design

Experimental design was appropriate.

Validity of the findings

With the inclusion of the ANOVA tables and the way the results have been revised, the conclusions appear to be valid. I have three minor questions.

Line 473 – Does this mean that low T men are more sensitive when offers are close to fair? Why would they discriminate between the ‘fair’ offer types less than men with high T?

Line 606 – why would this be? Does scoring high enough on the German political interest scale correlate with treatment of humans in economic contexts? Perhaps people interested in politics are more conscientious than others

Results – when you followed up the ANOVA with a post-hoc t-test, did you correct the p-value to account for multiple analyses? You mention you did this in line 516, but it is unclear whether than applies to all analyses, or only the one you reference in this paragraph.

Additional comments

147 – offers

176 – extraneous parenthesis

181 – You haven’t mentioned the use of photos yet, so this may be unclear. I would add something to the effect of: ..same number of new faces that were used to represent players in the UG.

212 – extra “at” – at home the morning of

379 – incorrect word – the structure of the…

415 – here you use player A and player B for the first time. Unless this is used later, I would just say proposer and respondent.

739 – correlation, not correlations

772, 801 – author names are both in text and in parentheses

Reviewer 2 ·

Basic reporting

no comment

Experimental design

no comment

Validity of the findings

no comment

Additional comments

the authors have done a good job addressing my concerns. I applaud the authors for making their data freely available. The paper makes a nice contribution to the literature.

---

## Round 0.3 · accepted · Accept

Thank you for responding to the few outstanding comments from the reviewers. It is my pleasure to accept your article for publication in PeerJ.